# A Review of Sustainable Pillars and their Fulfillment in Agriculture, Aquaculture, and Aquaponic Production

**Mark Schoor, Ana Patricia Arenas-Salazar, Irineo Torres-Pacheco, Ramón Gerardo Guevara-González** and **Enrique Rico-García** *

Department of Biosystem Engineering, Faculty of Engineering, Autonomous University of Queretaro, Querétaro P.O. Box 76010, Mexico; schoor.uaq@gmail.com (M.S.); arenas.uaq@gmail.com (A.P.A.-S.); irineo.torres@uaq.mx (I.T.-P.); ramon.guevara@uaq.mx (R.G.G.-G.)

* Correspondence: ricog@uaq.mx; Tel.: +52-4423644443

**Abstract:** Focusing on new food production methods and sustainable pillars' accomplishments has changed the definition of sustainable pillars themselves. Moreover, some general characteristics of the main pillars can be redefined in separate dimensions to better explain their positive sustainable impacts. Therefore, the main objective of this research is to redefine the sustainable pillars linked to food production and review the most important cultural and technological sustainability impacts they have, in addition to the three classic pillars: economic, social, and environmental sustainability. Cultural and technological sustainability are increasingly important complements to the traditional sustainability concept. Furthermore, new food production technologies and systems are influenced by ancient production methods, as well as by profitable crop selection. Traditional agricultural and aquaculture production in relation to more recent aquaponic production concepts are still a major part of global food security, but the better usage of waste materials or residues generates a more favorable agroecological impact. In conclusion, constantly redefining the sustainable pillars in the context of sustainable food production methods and proving the viability of their general production impacts is important.

**Keywords:** sustainability; food production; agroecology; responsibility; biodiversity





## 1. Introduction

The ecological impacts caused by human activities such as industrialization, urbanization processes, and the need for enlarged supply chains threaten our planet's sustainability and the environment [1]. Sustainability and the concept of sustainable development are related to industrial and agricultural production, which is integrated into environmental management. Both synonyms have different implications regarding potentials and limitations, according to their individual significance [2]. The three major areas of sustainability are the adjustments of economic, social, and environmental principles [3,4], which must coexist with each other [5]. In 2002, due to cultural disparity increases, culture was added as a fourth principle of sustainable development [6]. In addition to culture, technological advances and innovation are considered a new fifth pillar, due to the impact of technological improvements in sustainable development [7], which is mostly included in that of economic performance [8]. Furthermore, the objective of sustainable development is to support the balance between ecosystems and the economic exploitation of nature, by providing resources with a focus on preserving the Earth for future generations [9].

The long history of sustainability in the different production sectors has led to various policies and initiatives to address and integrate different disciplines, close sustainability gaps, and implement circular economy concepts [1]. Therefore, strategic economic and organizational activities must be adopted [10] to create a healthy environment, social cohesion, and economic efficiency [11]. The continual growth of the world's population

and the projected figure of 10 billion people living on our planet by 2050 necessitates transforming food production systems [12] to prevent environmental degradation and the rise of carbon emissions. Moreover, in many countries, there is little support for environmental sustainability considerations that permit an increase in economic growth and simultaneously protect the environment [13].

Food production systems represent a complex process involving food processing, packaging, distribution, selling, and consumption [14]. Furthermore, there are only a few organizations that control the food production sector and the inputs used, such as the required materials (seeds, agrochemicals, and machinery) that maintain agricultural production [15]. The transformation of food production systems must be improved and requires a redesign to conserve the environment and mitigate climate change [16].

This process includes globalizing food production [17], a model in which the implemented food production systems and methods [18] have a key role in achieving ecological sustainability in the application of innovative food production ideas [19]. In the past, there have been several investigations that, on the one hand, focused on sustainable production methods and the more efficient use of natural resources [20], mainly in the better exploitation of production spaces [21] and the better management of water in food production systems [12]. Moreover, there are complementary investigations focusing on declining agrochemical use [22] and the implementation of more agroecological production methods [23].

On the other hand, there is a new focus in the literature on enhancing food supply chains [24] and strengthening local food markets [25,26], which could provide organic foods in an innovative concept of "producing, buying, and eating locally" in food production [27]. Therefore, an increasing interest in backyard farming to complement partially local diets has been observed [28].

The greatest potential for implementing agroecological and sustainable food production exists in the better usage of waste and the residues originating from food production [12]. However, most investigations do not incorporate details of sustainability or do not demonstrate why a certain food production system is sustainable. This review focuses on an application of the different sustainable pillars to conduct an analysis on food production in aquaponic systems, including the two main components (agriculture and aquaculture) of aquaponics that could play a major role in accomplishing food security.

## 2. Literature Research Methodology

The initial literature survey in June 2022 was conducted to establish basic information about the concept of sustainability in relation to food production, which provided the keywords for the review. The research was performed by searching the MDPI, Elsevier, IEEE, Wiley, Taylor & Francis, and Google Scholar databases, concentrating the research on sustainability dimensions and food production systems. This differentiation helped to define a methodology with two main sections, the first analyzing the need to complement the traditional pillars of sustainability in food production and the impacts of cultural heritage and technological production, and the second examining the way that these predefined sustainable pillars are reflected in different food production systems.

Therefore, we performed a systematic review of the three traditional pillars of sustainability—economic, social, and environmental—to analyze the different concepts and implement the following sub-questions:

1. What are the specifics of the traditional sustainable dimensions?
2. Is there a need to amplify this definition, based on food production systems?
3. How sustainable are traditional food production systems (agriculture, aquaculture)?
4. How sustainable is an aquaponic production system in comparison to traditional production systems?

We attempt to answer these sub-questions by: (1) describing the traditional dimensions of sustainability; (2) linking traditional production systems to cultural practices and technology according to the local context, to ensure a better understanding of sustainable food production; (3) analyzing two traditional food production systems, agriculture and

aquaculture, and establishing their sustainable impact in food production, comparing these systems (4) to sustainable aquaponic food production.

Analysis of the traditional sustainable dimensions was performed by reviewing the initial concept of sustainability and its adoption by international institutions during the last few decades. Moreover, our analysis of the two traditional production systems (agriculture and aquaculture) showed that there are global cultural and technological differences that need to be considered when defining sustainable food production. Furthermore, food production was analyzed on the base of external influences which substituted the traditional production methods, as in the case of aquaponic food production.

Although aquaponics is categorized as a sustainable production method, we intend to compare the sustainability of aquaponic food production with the traditional production systems because aquaponics is a combination of agriculture and aquaculture, and both systems must surmount some difficulties to accomplish full sustainability.

This research was performed in three different languages, namely, English, German, and Spanish, to gain access to different information and studies about the pillars of sustainability. Moreover, conducting research in different languages helped us to review articles from different countries and to understand the cultural heritage of food production and the relevant applied technology in different parts of the world.

For this paper, we utilized different types of articles (reviews and scientific studies) and book chapters from different areas wherein sustainable pillars are applied and defined, on the topic of sustainability in food production systems. In general, the information is dated from 2018 to 2023; only established concepts and basic information about the topic date from before 2018.

## 3. The Concept of Sustainability and the Sustainable Pillars

The concept of sustainability or sustainable development is based on the German word "Nachhaltigkeit", which was defined in the book *Sylvicultura Oeconomica*, by von Carlowitz [29]. Implementing sustainability can be traced back to the year 1713, when it was used in relation to the forest industry [30], along with an implemented discussion of whether a forest can recover from humans' wood consumption [29]. Furthermore, it discusses the main principles of sustainability and how to strike a balance between resource consumption and the natural regeneration process of nature to ensure the survival of future generations [31]. The von Carlowitz concept was redefined several times, one of these being the Brundtland report in 1987, also known as "Our common future" [32]. The Brundtland report focused on sustainability's three main sections: the environment, the economy, and society (equity) [33]; more particularly, it directs us to our population's economic growth and the life cycle of our goods [34].

The United Nations redefined sustainability and sustainable development by implementing the 17 sustainable development goals and set the objective of accomplishing peace and prosperity by 2030 [35]. Furthermore, the United Nations implemented an agenda to support the preservation of the environment for future generations, implementing measures against climate change and reducing global warming [36] by adopting clean energy concepts [37]. Therefore, the following section concentrates on the three principal sustainability pillars, their classification, and the importance of the definition of the cultural and technological pillars of sustainability (Figure 1). The standard concept of sustainability, defined by Elkington, specified three main dimensions, namely, the economic dimension, the social and human dimension, and the environmental dimension [38], as stated in the Brundtland report [3]. Nowadays, these three dimensions [39] must be complemented by two new dimensions, these being cultural [40] and technological sustainability (Table 1) [41]. Therefore, employing the right indicators to measure the differences between regional geography, local problems, and population structure is important [39].

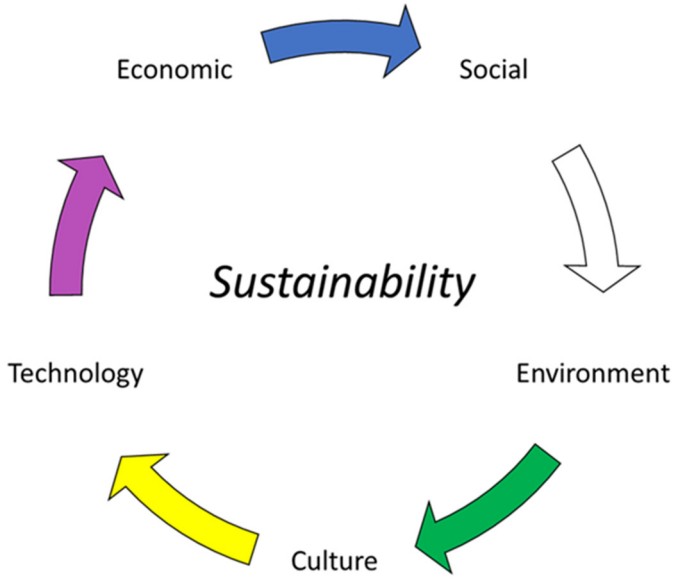

**Figure 1.** The five sustainable pillars.

**Table 1.** The sustainable pillars' characteristics.

| Pillar | Characteristics |
|---|---|
| Economic | - Minimize the ecological impact of economic activities<br>- Achieve economic and environmental balance<br>- Meet the need for innovative green products<br>- Assure nature's regenerative ability with economic progress |
| Social | - Satisfy basic human needs<br>- Prevent social injustices<br>- Achieve social responsibility<br>- Ensure educational development |
| Environment | - Maintain natural productivity<br>- Protect the environment<br>- Create alternatives to conventional resources<br>- Use green technology |
| Culture | - Represent cultural heritage and diversity<br>- Understand regional populational values<br>- Protect cultural heritage properties<br>- Share cultural values<br>- Achieve global cultural exchange |
| Technology | - Life cycle assessment<br>- Encourage the progress of digital technology<br>- Use environmentally friendly technological methods<br>- Optimize resource usage<br>- Use recycled materials in production processes |

*3.1. Economic Sustainability*

3.1.1. Generalities

The economic pillar of sustainability is defined as the balance between the environmental and social aspects in relation to economic goals [3], which can include the limitations that the human population must place on economic growth [42] to minimize the economic activities' ecological impacts [43]. The past has shown us that technological improvements have led to products with shorter life cycles [44], made without considering an economic

and environmental balance, in addition to the difficulty of establishing sustainable development [45].

In general, economic sustainability is more influenced by the social pillar of sustainability [42] due to the consumer decisions that are taken in the acquisition and the resulting quality of products and goods [46]. Therefore, the concept of consumer satisfaction with the least environmental impact possible must be adopted by introducing innovative products and goods [44] that have been produced according to environmental standards [47].

### 3.1.2. Economic Sustainable Food Production and Consumption

Social pressure from consumers can lead to a new focus on green product innovations [48]; economically, this can create corporate advantages and business benefits for companies investing in green products and process innovations [49], and, consequently, generate positive and sustainable economic growth for innovative companies [50]. There is a need for balancing all the long-term costs and benefits of economic activities to achieve sustainable economic growth, while simultaneously assuring nature's regenerative ability along with economic progress [42].

Nature's regenerative ability is even more important in the food production sector because of the adverse effects of climate change and the loss of productive croplands caused by rising food demand from the world's growing population [51]. Economic growth in agriculture depends on increasing crop yields and using non-conventional plants to provide food security [52]. There is high economic potential in the use of new technologies in saline areas that use salt-resistant plants, which can sustain crop productivity and reduce freshwater consumption. In addition, the new focus must be on plants with high nutritional values that are also resistant to soil contamination and can be implemented in industrial production with biotechnological functions [53]. Furthermore, the use of waste products from food production (organic material and water residues) to create a second product or animal feed becomes ever more important for establishing a sustainable cycle of production [54].

### 3.2. Social Sustainability

### 3.2.1. Generalities

The social sustainability pillar implies the continuation of satisfying basic human needs [42], which can also involve preventing social injustices such as child labor, inequality, and the abuse of working time limits and living conditions for workers in important manufacturing sectors inside international supply chains [55]. Moreover, social sustainability is connected to social responsibility and sustainable development which ensures economic substance without affecting the ecological environment [39]. This also includes the legal, ethical, and philanthropic obligations and responsibilities that the different members of society have [56].

Legal responsibilities include activities in line with laws and regulations, combined with ethical responsibilities, which involve moral conduct and behavior according to society's value system [57]. Therefore, social sustainability values include working rights, gender equality, non-discrimination, health coverage, and access to medical attention [58]. In addition, education was established as an important factor in the sustainable development goals outlined in the 2030 agenda of the United Nations, part of the social sustainability dimension and focusing on education's contribution to reaching economic and social wealth, which can also lead to a greater consensus on the importance of environmentally friendly practices [59,60].

### 3.2.2. Social Responsibility in Food Production

Even though social sustainability can be considered as being secondary [61], difficult to quantify the impact of [62], and often overlooked, attention is still paid to social practices and the relationship between economic growth and social responsibility [55]. In agricultural food production processes and due to the harmful environmental impact of using intensive

monoculture systems, population growth has created a high social demand for suitable practices [63]. Social damage and the loss of economic growth with intensive agriculture necessitate implementing sustainable strategies of cropland use and management practices, which include improving rural communities' long-term quality of life [60].

The life quality of rural communities also depends on socio-economic indicators that focus on the rural population's present conditions and families' specific incomes in rural areas due to agricultural production, labor availability, seasonal climate variations, and crop requirements [58]. Furthermore, the indicators reflect the farmers' actual conditions and regional development in the context of sustainable performance [64]. Rural regions and their local vitality create a state of isolation from urban areas, due to their lack of social policies [65]. Moreover, the rural population has different occupations and has passed through different educational systems [58], which necessitates a special regional social infrastructure to support local labor and discourage the local population from migrating [66]. The implementation of modern communication technology could also improve social cohesion, innovation, and education [67] in rural areas and stop regional depopulation [66].

### *3.3. Environmental Sustainability*
### 3.3.1. Generalities

In recent years, environmental sustainability has earned more recognition as a way to guarantee socio-economic sustainability and a healthy ecological environment [68]. The environmental sustainability pillar refers to maintaining natural productivity and ecosystem performance, in addition to protecting different wild species and preserving biodiversity [42]. Protection of the environment also includes the loss of natural wildlife, soil degradation, climate change, contamination in urban areas, and the intensification of agricultural production due to the incremental human consumption of natural resources [69]. The human consumption and production activities that cause environmental degradation are reflected in biodiversity loss, increasing natural hazards, food security, and human health effects [70].

An investigation of the contamination caused by an area's human population requires implementing an ecological footprint to control economic production density and development and the effects of urbanization [68]. Therefore, environmental sustainability necessitates improving and developing alternatives for conventional resources and using greener, cleaner, and more renewable processes [71] with a more balanced ecological footprint [68].

### 3.3.2. Environmentally Friendly Food Production

An important example of this includes the main effects and health risks caused by the use of chemical products, which have a negative effect on climate change and necessitate reducing industrial pollution to save the environment [69]. Especially in the energy sector, there is high potential for implementing sustainable energy requirements, such as by developing biofuels or using nanotechnology [71].

Focusing on food production, agricultural intensification is related to non-sustainable practices due to soil contamination, erosion, and water pollution, which cause environmental degradation and biodiversity loss [72,73]. Moreover, intensive agricultural production is the main cause of soil erosion because of salinization and a deficiency in organic substances, which cause biodiversity loss and general ill effects in agricultural regions [58]. Therefore, implementing sustainable agricultural practices and newly designed agricultural methods could play a key role in the development of sustainability and adapting practices to challenge the impacts of climate change [74].

### *3.4. Cultural Sustainability*
### 3.4.1. Generalities

In most analyses of the sustainability dimensions, the cultural pillar is included in the social dimension, in the form of the socio-cultural pillar [75], or under the cultural

diversity concept [58], which is not considered as important as the three original sustainable development pillars [76]. Nowadays, it is valuable to distinguish between social and cultural pillars, which might improve the alignment between external sustainability goals and organizational missions [77].

Culture and cultural heritage represent the diversity and representation of communities globally [78]. Moreover, protecting cultural heritage properties and the trespassing of values and meanings among generations is vital for implementing cultural sustainability [79]. Additionally, implementing cultural sustainability can help us to understand regional population values and implement the necessary changes in accordance with the general sustainability concept [76]. Today, cultural transmission is more possible and is different from the traditional cultural structures [80], due to the connection between people using social media and the possibility of international travel, in this way passing cultural information and heritage on to other countries with the use of multi-language functionality [77], implementing the idea of a global cultural village [78]. Social networks are also a reflection of cultural heritage, exposing the economic status and sustainable problems inherent in the concept of cultural globalization [81]. The assessment of culture has a key role in sustainable development, including the topics of traditions, vitality, economic capability, diversity, locality, and the ecological resilience of cultures and civilizations [77].

### 3.4.2. Cultural Heritage in Food Production

In agricultural food production, evaluating the cultural pillar is based on qualitative measurements, whereby rural community-based indicators are centered on self-evaluation via surveys and interviews with the rural population [58]. Ilieva et al. [82] have shown that there is a certain cultural benefit of agriculture due to community commitment, economic opportunities, and educational benefits. Moreover, implementing food production areas has fomented investigations into new agricultural technology [83] and increased awareness among the population regarding the benefits of ecological farming in contrast to conventional agricultural food production [82]. Agricultural food production is an important part of cultural identity and diversity that corresponds with food production methods, education, and providing food for the community [84].

### *3.5. Technological Sustainability*

### 3.5.1. Generalities

Technological sustainability is considered to be part of the economic pillar in most definitions, but technology now impacts all three pillars in the classic sustainable pillars definition [85]. The development and implementation of new technologies in shorter time frames have elevated natural resource consumption and decreased product life cycles, which will heavily impact future generations [44]. Therefore, it is necessary to have more efficient recycling and technology reuse processes [86], although this is difficult due to the high demand for new products [87].

There was a significant increase in the shift toward socio-technical transition [88] and an increased commitment to more scientific investigations based on sustainable methods [89]. Moreover, digital technological progress has transformed human society and the economic processes of technological sustainable development [90].

Technological innovations can help to achieve sustainability in various sectors and areas; in general, there must be a positive environmental impact [91]. The most recent investigations were focused on environmentally friendly technologies that resulted in sustainable accomplishments [92]. Therefore, technological advances can help to implement environmentally friendly practices and optimize resource usage [93], which are two important indicators of technological sustainability. This type of business model uses clean production practices, green innovation, and short supply chains as an advantage to achieve sustainable performance [91].

Modern technology concepts produced under sustainable methods include the use of recycled materials in the production processes of new goods [94], thus optimizing the

production process by integrating sustainable resources [95]. Moreover, new technologies that support sustainability [96] need to avoid contamination [97], especially regarding transportation distances in international supply chains, and protect social values and codes [98], which, in the end, could support regenerating the environment and biodiversity [99].

### 3.5.2. Sustainable Technology Innovation in Food Production

Agricultural methods have adapted to new technologies that permit high crop yield rates using improved machinery and developed enhanced genetic seeds and agrochemical products [58]; they have focused more on food security [100] and less on sustainability. One of the latest technological developments is the use of nanotechnology in the production of biofuels [71] or agriculture [101].

In agriculture, sustainable technology can be used for optimizing production processes and transforming food production practices into green and clean methods [102]. For example, the treatment of wastewater from food production processes can help achieve sustainability by using new and innovative technology [103]. Moreover, wastewater treatment technologies and economic, environmental, and energy usage rate analyses [104] can help producers to implement technologies such as aquaponics [105].

## 4. Sustainability in Food Production

Human population growth and the global scale of urbanization [106] necessitate establishing expanded food production and highlight the importance of food supply chains from rural to urban areas [107]. Moreover, transportation from rural to urban areas has a great environmental impact [108] so it requires more local sustainable production methods [109]. Therefore, to understand sustainability, analyzing the different sectors of food production where sustainability has been applied is important. This section of the article focuses on food production in aquaponic systems, looking at two components of the system: agriculture and aquaculture production.

### 4.1. Sustainability in Agriculture Food Production
#### 4.1.1. Recent Agricultural Food Production

Agriculture food production provides economic, social, cultural, and technological benefits to the human population [110] and, in the different stages of planting, growing, harvesting, and transporting the final product, offers labor opportunities to more than one billion people globally [111]. Moreover, increasing alimentation needs and breaking off from nutrient accumulation have impacted the balance between food demand and supply, which will bring new difficulties for the food production sector in the coming years [112].

The consequences of higher world food demand included an increase in synthetic fertilizer use, causing massive environmental impacts due to toxic soil pollution and changes in the physicochemical soil conditions [113], which has led to general soil degradation, with long-term effects on agricultural productivity and human health [112]. Furthermore, recent studies about contamination showed negative effects caused by the use of more than one agrochemical pesticide on 64% of global agricultural land. Moreover, in addition to 64% of affected soil, 31% of agricultural soil is considered at high risk of contamination [114]; therefore, more environmentally friendly methods are required [115].

There is a significant contrast between the different concepts of agriculture. Intensified agriculture is the most common concept, which can be understood as productivity increases in the same cropland space [116,117]. Food production intensification is highly dependent on advances in agrochemical and genetic technologies, but causes soil and salinity degradation and soil sterility, causing the soil to no longer be viable for agricultural food production [118]. Nowadays, the development of new agro-technologies is fundamental to meeting the rising global demand for food products [119] and simultaneously increasing food production, with less environmental pressure [120]. The key to more environmentally friendly food production is sustainability and the use of more renewable resources with lower ecological impacts [115]. In sustainable agricultural development,

there are some patterns that must be addressed, such as the conservation of agriculture, the perspective of organic farming, the use of biological agrochemicals, and better pest management [120], with the focus being on a sustainable production system with lower environmental, economic, and social impacts [121].

### 4.1.2. Impact of the Sustainable Pillars in Agriculture

The search for more sustainable production methods must involve food production systems adopting resource-saving and environment-conserving concepts, along with efficient production, to secure market competitiveness; as a result of this practice, farmers' living standards will improve [122]. According to the authors of [123], the implementation of sustainable production methods involves working with natural and social systems, including conserving agriculture in general, and the agroecological process and food production process principles more specifically. The major principle of sustainable food production and the reduction of environmental impacts [124] must take into account soil management, crop management, and genetic management to protect biodiversity as part of the concept of agricultural conservation [125].

Moreover, soil management is vital for food production due to the provision of nutrients for agricultural plant growth, with the essential soil nutrients for plants being N, P, and K [126]. Furthermore, there is a difference between qualitative and quantitative production techniques [58]. Qualitative techniques are based on visual evaluations of soil structure and texture [127] or of quality and fertility, which impact cropland management [128]. Moreover, quantitative techniques suggest compiling data-based literature and creating empirical models of food production [129].

Food production sustainability is at risk in plant nutrition because excessive chemical fertilizer usage and the consequent soil degradation require considering physical soil condition rehabilitation [130]. Moreover, crop rotation, biofertilizers, and precision agriculture can help to increase crop productivity and prevent cropland from decreasing in soil fertility [120].

Therefore, to measure sustainability in agricultural practices, it is important to adopt viable three-dimensional indicators to establish the different farming systems, methods, and local considerations [58]. These environmental indicators of agricultural production consider the cause-and-effect relationship between food production and the environment, including agricultural biodiversity, agrochemicals use, water usage, and soil composition [131]. The indicators focusing on economic considerations incorporate the viability, profitability, stability, liquidity, and productivity of the inputs and outputs of agricultural cropland production [58]. The social indicators are based on the farming methods of the community and community wellbeing [60] in relation to agricultural sustainable practices according to the local ecosystem [132].

### 4.2. Sustainability in Aquaculture Food Production

#### 4.2.1. Industrial Aquaculture Production

Besides agricultural plant-based food production, a second common system is aquaculture production, which can be defined as human participation in the farming of aquatic organisms by adapting freshwater and marine habitats [133]. Aquaculture food production is the quickest-growing food production sector, using aquatic food production systems in inland locations or open sea marine systems [134], with the main regions in Asia practicing aquaculture for food production [133].

There are different aquaculture production systems; the most common are cage, flood-plain, net-pen, pond, raft and long-line aquaculture and raceway farming, recirculating aquaculture systems, and rice–fish farming [133]. Aquaculture production can be classified as intensive, semi-intensive, or extensive production [135], cultivating aquatic animals via monoculture, polyculture, or integrated fish farming [136]. Globally, there are more than 622 aquatic animal species reported that can be farmed in aquaculture food production

systems, the most common of these being carp, catfish, shrimp, salmon, and tilapia [133] due to their trade values [134].

Aquaculture farmers are greatly concerned with the elevated costs of raw materials and equipment requirements [137]. On the one hand, using industrial aquafeed aliments creates a large environmental impact due to different types of contamination (water and mud sediments) [138]. On the other hand, the high levels of use of hydric resources and the return of wastewater to the environment causes aquaculture food production's sustainability to be questionable [139].

### 4.2.2. Sustainable Optimization Potential of Aquaculture

The new sustainable aquaculture food production methods demand the implementation of strategies that consider the effects of climate change, environmental risks, biosecurity, and the health of aquatic species while applying new technologies to guarantee productivity [137]. There is a high potential to solve one fundamental problem of aquaculture food production, namely, the use of fish meal protein for aquatic animal feed, which is affecting the system's sustainability [140]. These types of aliments have costly production processes due to the high demand for raw materials, causing several environmental issues [138].

A viable sustainable alternative for aquatic feed aliment involves using black soldier fly larvae [141], which are rich in protein [140]. Even more viable is producing and using cheaper, locally produced nutritious ingredients for aquafeed, such as banana and cassava peels or rice and maize brans as carbon sources inside the food supply chain [137], contributing to the circular economy with efficient waste use [142].

Aquaculture is a water-intensive food production system, with a high usage of freshwater [143], which creates a high water footprint for aquatic animal production [144]. Furthermore, aquaculture food production has a high environmental impact, due to the hydric discharge of wastewater concentrations of organic, inorganic, and microalgae into the environment [145]; without a nutrient eradication process, this causes the eutrophication of natural aquifers [146]. Due to the resulting nitrogen pollution in the water, aquaculture production generates a greywater footprint, with the concentration being dependent on the cultivated aquatic animal species [147].

Even though nitrogen is a vital nutrient in agricultural food production, the nitrogen concentration from aquacultural food production, as seen in pond sludge and wastewater discharge, has damaging effects on the local ecosystems; therefore, more sustainable aquacultural production practices and nitrogen-efficient usage are required [146]. One method to reduce aquaculture sludge waste's environmental impact involves using black soldier fly larvae and biologically transforming the sludge waste, which is cost-efficient and easy to handle in comparison with traditional waste-reducing technologies [148].

To be more sustainable, aquaculture needs to consider the nature input that is required for food production, depending on the farmed species, and possibly increasing production to its maximum potential [149]. Moreover, integrating technological advances to improve resource usage efficiency and energy efficiency, along with recirculating aquaculture systems such as aquaponics, can not only improve food security but also decrease aquaculture production's environmental impact [137].

### 4.3. Sustainability of Aquaponic Systems

#### 4.3.1. Aquaponic System Development

Aquaponics is a technology that can support a short food supply chain and has the capability to be installed in smaller spaces [107]. Aquaponic systems combine agricultural plant production, with the use of plant cultivation via hydroponics, and aquaculture fish production in one incorporated system [150]. There are two different designs of aquaponic systems; the most common is the recirculating system [105], in which nutrient-rich water resources from fish production run in a circle-loop layout between the fish culture area and the plant production bed [151]. Less well-known is an open aquaponic system with combined soil-based plant cultivation, which consists of an aquaculture production system

using aquatic wastewater for soil irrigation to foment plant production [152]; this system is also known as a wastewater irrigation system [153,154].

Therefore, due to the beneficial connection between fish and plant production, aquaponics is considered a sustainable production method [155]. The produced aquatic waste intended for fostering plant growth [156] uses fish defecation as nutritious fertilizer [157]. Furthermore, the natural nutrients concentrated in aquatic wastewater can limit chemical fertilizer and freshwater use due to a second exploitation of hydric resources, which has a positive environmental impact [158].

### 4.3.2. Wastewater Use in Aquaponic Food Production

Wastewater use in aquaponic systems reduces the need for external nutrient input and waste disposal [159] due to water reuse. Aquaponics adopts the principles of the circular economy, which consist of reducing resource consumption and discharge into the environment [160]. Moreover, circular economic enhancement and the optimization of food production [161] can reduce the waste generation caused by food production [160]. Reusing aquatic animal wastewater could involve implementing a zero-waste process [162], with water savings of from 80% up to 90% in comparison with traditional production methods [163]. Therefore, the water efficiency of aquaponic systems is higher than the conventional techniques and reduces chemical fertilizer use [150,164].

The responsible use of resources and production efficiency make aquaponic systems appropriate for educational programs, with a positive social outcome [155]. Additionally, aquaponics systems demand rigorous monitoring, control, and management, which can be enhanced using technology based on the Internet of Things (IoT) to optimize water use and food production [156].

Moreover, aquaponic systems have an important role in terms of providing food security [165,166] because they can be implemented regionally, offering the production of various vegetables, fruits, and aquatic animals near the consumers' homes [167]. Regional food production and local backyard production using aquaponics can support sustainability [168] by lowering food transportation and $CO_2$ contamination [159].

Aquaponics, as a type of urban agriculture ensuring the supply of fresh food [124], also contributes to the local economy by creating green jobs and offering social integration [169]. Furthermore, aquaponics enables urban areas to be more sustainable by promoting self-sufficiency and the implementation of low-carbon food production by growing high-value plants [170]. Additionally, aquaponics presents a valuable market opportunity due to the new ideology of buying local [26], environmentally friendly, and organic products, which are free of antibiotics and agrochemicals [159,171]. Therefore, the low water usage and wastewater nutrient use, in combination with design variety and simple adaption to different locations, production types, and climate conditions are strong arguments for aquaponics to be considered as a sustainable production method [169] that replicates natural systems by farming aquatic animals and plants in symbiosis [150].

### 5. Discussion

Global food production depends on adopting sustainable production methods because of the overexploitation of fertile soil, cropland loss [172], high agrochemical and water usage in agriculture [173] and aquaculture systems [174], and water contamination. This requires a variety of optimizations to transform the different systems into sustainable ones [175,176]. Agricultural production is reaching its limit due to intensification [177], to the point that using even more agrochemicals does not promise more productivity [158]. Furthermore, agricultural food production is reverting to organic food production [178,179], with less substantial yield rates, but the system is more environmentally friendly and there is more plant diversity due to the use of multi-cropping or polyculture plant production systems [180,181]. Moreover, aquaculture wastewater discharges into close aquifers are also creating certain negative environmental impacts [182], are wasted, or are not used in alternative food production as a valuable nutritious asset [183].

Aquaponics, combining agriculture and aquaculture in one system [184], is a viable alternative food production system that is regarded as sustainable due to the symbiotic growth of aquatic animals and plants [164] and the use of nutrient concentrations in the wastewater [156]. In fact, an aquaponic system's water efficiency is higher than in standard agriculture and aquaculture and it prevents agrochemical use [150], which means that aquaponics is an eco-friendly and cost-effective system [164]. Moreover, aquaponics presents an opportunity to reduce production spaces by using closed recirculating systems [105] or an open soil-based irrigation system [152].

Yep and Zheng [150] mention that aquaponics is a cleaner and more sustainable food production practice than conventional food production in general, but suggest that aquaponics is close to being fully sustainable and accomplishing all the sustainability pillars. However, sustainable development depends on local conditions and national policies to achieve full sustainability [66].

In aquaponic systems, the impact of technological sustainability when using an innovative production method is important [159]. Therefore, demonstrating aquaponics' economic impact and the cost-effectiveness of the different production scales (backyard cultivation, small-scale farming, rural farming, and industrial farming) is important [164]. Furthermore, the social and educational impacts are difficult to measure, but there could be further investigations regarding the handling of agrochemicals in food production and the development of a social conscience about protecting the environment and biodiversity [12]. There is a positive environmental impact due to the better use of water when producing two proteins, namely, fish and plants, in one system but there must be further investigation of aquaponics' long-term impact, due to the use of different types of plastics and metals, energy use in recirculating systems, and the organic waste materials of the production processes [185]. The most difficult sustainable pillar to accomplish is cultural sustainability because only a few countries and regions have an ancient heritage and traditions of hydroponic production or early aquaponic production, such as the Chinampas food production in Mexico [186].

All these considerations should be applicable to agriculture and aquaculture and help to improve the impact of sustainable food production in traditional production systems. In addition, farmers and agricultural regions need help from different institutions (governments, universities, and non-profit organizations) to develop new sustainable practices based on technological advancements [187]. For example, there is a high potential in terms of using organic fertilizer, waste management [188], and digitalization [189], which could be facilitated by new institutional sustainable strategies.

In the future, there must be more investigation into agricultural, aquacultural, and aquaponic food production to ensure sustainability's being fully applied to accomplish the different pillars, with a focus on food security, lowering environmental impacts, and high product quality, while respecting farmers' cultural food heritages and remaining accessible for the whole of social society. Moreover, there must be investigations and cultural exchanges about production methods, systems, and practices, as well as if they are viable for other countries and regions.

## 6. Conclusions

In the future, it will be important to concentrate scientific investigations more closely on sustainable food production systems that fulfill the five established sustainability pillars (economic, social, environmental, cultural, and technological). Therefore, our investigations must be directed toward renovating the established food production systems (agriculture, aquaculture, and livestock) or combining different food production systems (aquaponics) to detect their hidden potential for optimizing the system. The new focus must concentrate on accomplishing the sustainability pillars and goals. Moreover, cultural and technological sustainability must be more integrated into the concept of sustainability, to help to understand the importance of the acceptance and handling of new food production systems.

Aquaponics may be an alternative system that has the potential to be a sustainable method of food production, but there are still some questions about its sustainability, due to the use of non-organic materials, energy usage (water pumps), waste, and water usage. New and innovative food production systems are important for the future of our alimentation, but there must be long-term investigations about each system's sustainable impact and in which category (backyard, small-scale, and industrial) production is viable.

In addition to alternative production systems, it is important to verify the traditional production systems, or a combination of new and innovative systems with traditional systems and determine whether increasing each system's sustainability is possible. In traditional systems, there is still hidden potential in using organic biorational fertilizers or pesticides, different soil accommodations, or crop combinations (multi-cropping), and, in traditional aquaculture, the better usage of wastewater, ponds, and fish food on a renewable organic basis (with black soldier fly larvae).

**Author Contributions:** Investigation, M.S. and A.P.A.-S.; writing—original draft preparation, M.S. and A.P.A.-S.; writing—review and editing, M.S., A.P.A.-S., E.R.-G., I.T.-P. and R.G.G.-G. All authors have read and agreed to the published version of the manuscript.

**Funding:** This research was financed by the student investigation support programe FOPER of the Autonomous University of Queretaro with the funding number FOPER-2021-FIN02527.

**Institutional Review Board Statement:** The study was approved by the Ethics Committee of Universidad Autónoma de Querétaro (protocol code CEAIFI-XXX-XXX-TI).

**Informed Consent Statement:** Not applicable.

**Data Availability Statement:** Not applicable.

**Acknowledgments:** The authors acknowledge the financial support from El Consejo Nacional de Ciencia y Tecnología (CONACYT).

**Conflicts of Interest:** The authors declare no conflict of interest.

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
