# Peer review of "A Review of Sustainable Pillars and their Fulfillment in Agriculture, Aquaculture, and Aquaponic Production"

_sustainability, doi:10.3390/su15097638_

Round 1

Reviewer 1 Report

Comments to the authors:

The present study describes the sustainability pillars for food production in order to improve the sustainable food production and food security. The authors have proposed five sustainability pillars, however, which is not focused and well defined. It needs to be presented in a better manner to understand it. The pillars are written in general state, which is not written in depth to make the audience aware about it. In my opinion, ‘technology’ is a part of economic pillar, while ‘culture’ is a part of social pillar of the sustainability. Thus, both are connected with the environment pillar of the sustainability.

The manuscript has many errors, in many places it is hard to understand due to poor language, long sentences and lacks in connectivity (synchronization). Hence, there is a scope to improve the quality of the Ms to make it readable to a wide range of audience. The Ms lack the coherence in the text and lacks in the connectivity of the proposed sustainability pillars.

The conclusion section needs to be reduced. The conclusion and discussion section needs to be written precisely. I regret to inform you that I cannot accept the article in its current form.

Specific comments:

Title- The title of the article needs to be updated.

Abstract- The abstract portion needs to be re-written.

Introduction- The introduction section needs to be improved because this portion is not focused.

Results- The results section needs to be improved with a coherence and connectivity.

Discussion- The discussion part is not written well, which needs to be written well to describe the connectivity and importance of the study.

Conclusion- This part also need to improve.

Reference- Too much references.

I have also given the comments in the annotated document, so check it out and address it correctly.

The language of the article needs to be improved.

Author Response

We are grateful for all comments made for reviewers.

Related comments for reviewer 1 were highlighted in yellow in the MS.

Reviewer 1

* The present study describes the sustainability pillars for food production in order to improve the sustainable food production and food security. The authors have proposed five sustainability pillars, however, which is not focused and well defined. It needs to be presented in a better manner to understand it. The pillars are written in general state, which is not written in depth to make the audience aware about it. In my opinion, ‘technology’ is a part of economic pillar, while ‘culture’ is a part of social pillar of the sustainability. Thus, both are connected with the environment pillar of the sustainability.

Thank you very much for the comments.

During our research we detected that a variety of food production systems are declared to be sustainable, but without an explanation why they are considered to be sustainable. Therefore, we analyzed the traditional dimensions of sustainability; economic, social and environmental. Food production is based on technology and on innovation which could be considered as an economic factor, but the past has shown us that not all food production system based on high technology are sustainable, for example, a lot of the technologies implemented during the green revolution are not sustainable. Therefore, we implemented the technological pillar of sustainability and separated from the economic pillar to show that there can be sustainable technology in food production, but maybe it is not as profitable as monoculture food production systems. Moreover, food production is based on cultural heritage and practices which were implemented centuries ago. Social sustainability is based on the educational impact of food production and the well being a farmer could gain by using sustainable production methods. Social sustainability does not reflect the cultural aspect of a regional food production system, the use of regional grains and regional adapted production systems. Therefore, we think it is really important to separate social and cultural sustainability to analyze different food production systems, because, due to the green revolution, a lot of farmers use imported technology and production knowledge instead of local adapted production systems.

* The manuscript has many errors, in many places it is hard to understand due to poor language, long sentences and lacks in connectivity (synchronization). Hence, there is a scope to improve the quality of the Ms to make it readable to a wide range of audience. The Ms lack the coherence in the text and lacks in the connectivity of the proposed sustainability pillars.

We have evaluated the situation of the English quality of our article and we have decided to send the article for further revision to native English speakers so that we can have a better English quality of our manuscript. Moreover, we made it more readable by preediting our content and connecting in a better way the different ideas.

* The conclusion section needs to be reduced. The conclusion and discussion section needs to be written precisely.

We think that our conclusion and discussion have an accurate size and gives the reader the context about sustainability in food production. Also, we are discussing and concluding perspectives about sustainable food production.

* Title- The title of the article needs to be updated.

We think that the title “A Review of Sustainable pillars and the fulfilment in agriculture, aquaculture and aquaponic production” is accurate, because we are reviewing the traditional pillars of sustainability and adding two new pillars (culture and technology) to our definition. These new pillars are normally not considered and defined in food production system. Nevertheless, in the introduction we are describing the relation between sustainability and food production and giving an explanation of the relation between the food production systems and the sustainability pillars which are considered for this review article.

* Abstract- The abstract portion needs to be re-written.

The Abstract has been improved by a native speaker and adjusted to provide a better overview of the content of our article.

* Introduction- The introduction section needs to be improved because this portion is not focused.

The introduction is focused on the traditional pillars of sustainability and to understand the importance of culture and technology in food production, so the reader could notice that this two pillars must be analyzed separately.

* Results- The results section needs to be improved with a coherence and connectivity.

We based our review on the sustainable pillars that are viable in food production and are connected directly with food production. We think that our results are accurate, because in the first part we present our investigation about the sustainable pillars in food production and in the second part we explain three different systems, why they are sustainable and why not.

* Discussion- The discussion part is not written well, which needs to be written well to describe the connectivity and importance of the study.

We treated this topic above and we corrected the English.

* Conclusion- This part also need to improve.

We treated this topic above and we corrected the English.

* Reference- Too much references.

We made a deep analysis about sustainable pillars, therefore, we needed to review the literature directly with food production systems, but also close related literature, especially in the cause of the cultural and technological pillar. Culture and technology are not often considered to explain sustainability characteristics of food production which required further explanations. Moreover, in the last weeks there were many reviews published in Sustainability with the same number or even more references than our article has.

* Correction´s the reviewer marked yellow in the uploaded document (I have also given the comments in the annotated document, so check it out and address it correctly)

 We reviewed your document and made all the corrections you marked in the original manuscript. All changes are marked in yellow.

Reviewer 2 Report

There are notes in the file.

Author Response

We are grateful for all comments made for reviewers.

Related comments for reviewer 2 were highlighted in blue in the MS.

Reviewer 2

* The authors of the manuscript provided a very broad overview of sustainability pillars/dimensions and potential pillars/dimensions but did not mention any institutional pillar/dimension initiatives analyzed by other researchers (Spangenberg, J. H., Pfahl, S., Deller, K. (1999); Spangenberg, J. H. (2002); ÄŒiegis (2003); Djoghlaf A. (2007); Omar, O., Kamarulazizi, I., Kanayathu, K., Marlinah, M. (2015); Bencardino, M., Nesticò, A., Esposito, V., Valanzano, L. (2022); etc.). If the authors have a different opinion, they can provide arguments or refer to opposing researchers, but this is not even mentioned now.

Thank you very much for your comments.

We know that institutional sustainability is a very important topic, and it can also influence food production systems by the implementation of norms, laws, rules and advices by governments (federal, state, local), universities and ONG´s. In our review article we are concentrating on the sustainability of three food production systems; agriculture, aquaculture and aquaponics and how the production processes are taken out. We did not focus on the political and institutional environment where a food production is taking place, since these vary from region and countries. We concentrated on our experience with small agriculture and aquaculture farmers in rural areas of Mexico and searched for general and specific information to every system studied and analyzed the problems which were reported in scientific research articles.

In addition, we reviewed the authors you mentioned and find their positions interesting, but we think that it requires a deeper analysis of institutional sustainability and future studies related to food production systems and how institutional sustainability could help to implement good production practices to push food production sustainability even further. Therefore, we added a new paragraph to our discussion to show other investigators that there is a lot of potential for future investigation:

Original: “All these considerations could be also applicable for agriculture and aquaculture to estimate the sustainable impact of food production, in general.”

Modified version: “All these considerations could be applicable for agriculture and aquaculture and help to improve the sustainable impact of food production in traditional production systems. In addition, farmers and agricultural regions need help from different institu-tions (governments, universities, and non-profit organizations) to develop new sus-tainable practices based on technology advancements [187]. For example, there is a high potential in using organic fertilizer, waste management [188], and digitalization [189], which could be facilitated by new institutional sustainable strategies.”

Lines 561 to 567

* Manuscript title is "A Review of Sustainable pillars… ", so it should be improved a bit.

We think that the title “A Review of Sustainable pillars and the fulfilment in agriculture, aquaculture and aquaponic production” is accurate, because we are reviewing the traditional pillars of sustainability and adding two new pillars in culture and technology to our definition. These new pillars are normally not considered and defined in food production system. Nevertheless, in the introduction we are describing the relation of the sustainability and food production and giving an explanation of the relation between the food production systems and the sustainability pillars which are considered for this review article. Like we mentioned before, we know that institutional sustainability is important to measure part of sustainability, but the focus was in the traditional pillars and the direct related new pillars to food production systems, culture and technology, therefore, we like to keep our title as we had in the original version.

* Affiliations (in lines 6-9) of the authors are in Spain, but not in English.

We changed the affiliations from spanish to english

Original: “1            Department of Biosystem Engineering, Faculty of Engineering, Autonomous University of Queretaro; Concá, C.P. 76149, Mexico; [email protected]; [email protected]

2                 Department of Biosystem Engineering, Faculty of Engineering, Autonomous University of Queretaro, Querétaro C.P. 76010, Mexico; [email protected]; [email protected]; [email protected]

*                  Correspondence: [email protected]; Tel.: 52 4423644443”

Modified version: “1         Departamento de Biosistemas, Facultad de Ingeniería, Universidad Autónoma de Querétaro, Concá C.P. 76149, México; [email protected]; [email protected];

2                  Departamento de Biosistemas, Facultad de Ingeniería, Universidad Autónoma de Querétaro, Querétaro C.P. 76010, México; [email protected]; [email protected]; [email protected]

*                  Correspondence: [email protected]; Tel.: 52 4423644443”

* (Salvador et al. 2014) are in line 291, (Buscaroli et al. 2021) are in line 471, but the authors aren’t in the list of references.

We added Salvador et al. 2014 to the reference list and changed the mistake in line 291 to [104], which required also the actualization of the following references. Buscaroli et al. 2021 in line 471 was already in the references list and we added the references number to the the text [26].

* I would suggest to the authors briefly discuss the use of the term "aliment" in the manuscript.

We analyzed the use of the word aliment and when we are referring to human alimentation, we changed it to “food” and only use this term when we are referring to alimentation of animals.

* The list of references was prepared not under MDPI Instructions for Authors.

We changed the references to the MDPI format from the template document.

* Small corrections are needed (In Addition … (in line 159)

Original: “In Addition”

Modified: “In addition”

* Food production (in line 125); Food Production (in lines 164, 240, 280, 304); food production (in line 203) have differences in sub-titles

We corrected these observations and changed all to “food production” so every sub-title has the same writing.

* [82] have shown that there… (in line 243);

Original: “[82] have shown that there is certain cultural benefit of agriculture due to community commitment, economic opportunities, and educational benefits.”

Modified version: “Ilieva et al. [82] have shown that there is a certain cultural benefit of agriculture due to community commitment, economic opportunities, and educational benefits.”

* Therefore, technological advances con help… (in line 268)

Original: “Therefore, technological advances con help to implement environmentally friendly practices with the optimization of resource usage [93], being one of the most important indicators of technological sustainability the type of business, which uses clean pro-duction practices, green innovations, and short supply chains as advantage to reach sustainable performance [91].”

Modified version: “Therefore, technological advances can help to implement environmentally friendly practices and optimize resource usage [93], which are two important indicators of technological sustainability. This type of business uses clean production practices, green innovations, and short supply chains as an advantage to reach sustainable performance [91].”

* … [117].Nowadays… (in line 325)

Original: “[117].Nowadays”

Modified version: “[118]. Now”

* … [152, 153. (in line 436); etc.)

Original: “Less known is the open aquaponic system with a combined soil-based plant cultiva-tion, which consist in an aquaculture production system and the use of aquatic wastewater for soil irrigation to formant plant production [152], this system is also known as a wastewater irrigation system [152, 153.”

Modified version: “Less known is the open aquaponic system with a combined soil-based plant cultiva-tion, which consists of an aquaculture production system using aquatic wastewater for soil irrigation to foment plant production [152]; this system is also known as a wastewater irrigation system [153, 154].”

Author Response

We are grateful for all comments made for reviewers.

Related comments for reviewer 3 were highlighted in green in the MS.

Reviewer 3

* The manuscript “Review of sustainable pillars and the fulfilment in agriculture, aquaculture and aquaponic production” discusses on the present sustainable pillars and the present agricultural, e.g., agriculture, aquaculture and aquaponic production systems. The subject area of the manuscript is important in fulfilling the criteria in modern agricultural production technology. The manuscript is nicely narrated, however, there are many grammatical errors and mistakes. The manuscript is recommended to check thoroughly by native speaker. There are also some major issues that are required to solve before the manuscript accepted in Sustainability.

Thank you very much for your comments.

We have evaluated the situation of the English quality of our article and we have decided to send the article for further revision from native English speakers so that we can have a better English quality of our manuscript.

* Abstract: Line 22; “materials r residues....” should be “materials or residues....”

Original: “materials r residues”

Modified version: “materials or residues”

* Introduction: Line 33; mayor should be major...

Original: “mayor”

Modified version: “major”

* Introduction: Line 36; principal should be principle

Original: “principal”

Modified version: “principle”

* Introduction: Line 41; “exploitation of by nature”.....need correction.

Original: “Furthermore, the objective of sustainable development is the utility balance between the ecosystem and the economic exploitation of by nature provided resources with the focus on the preservation of our earth for future generations [9].”

Modified version: “Furthermore, sustainable development’s objective is using the balance between the ecosystem and the economic exploitation of nature by providing resources with a focus on preserving Earth for future generations [9].”

* Line 87; principals should be principles

Original: “principals”

Modified version: “principles”

* I suggest to add discussion with the main body (observation) rather placing in separate

paragraph.

We think the separation between discussion and the main body is appropriate, not only because it is adopted to the editors instructions of Sustainability, but also we are discussing to core topics of the main body, sustainable pillars and food production systems and by adding the discussion to the main body it could indicate we are only discussing food production systems.

Reviewer 4 Report

Dear authors

After reading carefully the manuscript, my main concern is that the authors do not explain how they have performed their review and it is important to describe this methodology. There is a lack of a methodological framework of how they have performed the particular review.

The research questions could have been more specific and refer to:

1. To identify the key elements of their research

2. Why we need to perform this study

3. Which are the benefits

4. Which are the problems or difficulties of performing this review

What type of sources/digital library: Scopus, Science direct, Springer, Taylor and Francis, IEEE digital library, etc, the authors have consulted?

Which were the inclusion Criteria during their review. Please see below some examples

• Research field

• Language English

• Publication date from…..

• Type of work scientific studies

• Availability full text

Which were the Exclusion criteria during the review. Please see below some examples

• Not belong to the field of research

• Non English

• Publication date before…..

• Non scientific studies

• Not related to the subject of interest

The authors need to perform also a detailed language editing, since sometimes Spanish words and phrases appear in the text.

Author Response

We are grateful for all comments made for reviewers.

Related comments for reviewer 3 were highlighted in grey in the MS.

Reviewer 4

* After reading carefully the manuscript, my main concern is that the authors do not explain how they have performed their review and it is important to describe this methodology. There is a lack of a methodological framework of how they have performed the particular review.

Thank you very much for your comments.

After your recommendation we added a methodological framework to our article.

Original: “none”

Modified version: “The initial literature research in June 2022 was performed to provide basic information about the sustainability concept in relation to food production, which provided the keywords for the review. The research was performed by searching the MDPI, Elsevier, IEEE, Wiley, Taylor & Francis, and Google Scholar databases, concentrating the research on the sustainability dimensions and food production systems. This differentiation helped to define a methodology with two main sections: analyzing the need to complement the traditional sustainability pillars in food production and the aspects of cultural heritage and technological production impacts, and reflecting these predefined sustainable pillars in different food production systems.

Therefore, we performed a systematic review of the three traditional pillars of sustainability—economic, social, and environmental—to analyze the different concepts and implement the following sub-questions:

  1. What are the details of the traditional sustainable dimensions?
  2. Is there a need to amplify this definition based on food production systems?
  3. How sustainable are the traditional food production systems (agriculture and aquaculture) are?
  4. How sustainable is an aquaponic production system in comparation to the traditional production systems?

We attempt to answer these sub-questions by: 1. describing the traditional sus-tainability dimensions; 2. based on the traditional production systems, we highlighted that culture and technology must be more specifically defined to better understand sustainable food production; 3. we analyzed two traditional food production systems, agriculture and aquaculture, and their sustainable impact in food production and compared these systems; and 4. then compared them to sustainable aquaponic food production. 

Although aquaponics is categorized as a sustainable production method, we want to compare aquaponic food production’s sustainability with the traditional production system, because aquaponics combines agriculture and aquaculture and both systems have some difficulties to overcome in accomplishing full sustainability.”

* The research questions could have been more specific and refer to: 1. To identify the key elements of their research; 2. Why we need to perform this study; 3. Which are the benefits; 4. Which are the problems or difficulties of performing this review.

We implemented our research questions after your recommendations and a review of examples for research questions implementation.

Original: “none”

Modified version: “1.     What are the details of the traditional sustainable dimensions?

  1. Is there a need to amplify this definition based on food production systems?
  2. How sustainable are the traditional food production systems (agriculture and aquaculture) are?
  3. How sustainable is an aquaponic production system in comparation to the traditional production systems?”

* What type of sources/digital library: Scopus, Science direct, Springer, Taylor and Francis, IEEE digital library, etc, the authors have consulted?

We attached the sources and the type of sources we used in our methodical framework.

Original: “none”

Modified version: “The research was performed by searching the MDPI, Elsevier, IEEE, Wiley, Taylor & Francis, and Google Scholar databases, concentrating the research on the sustainability dimensions and food production systems.”

* Which were the inclusion Criteria during their review. Please see below some examples: Research field, Language English, Publication date from….., Type of work scientific studies, Availability full text.

Our main research field was sustainable food production, but divided into to three main areas; agriculture, aquaculture and aquaponics. Therefore, our main focus was on these production systems, but in addition we made an open investigation about different sustainable which are not considered in food production and are suitable to define better sustainability of food production. Since there is very view information about cultural and technological sustainability in direct food production, we opened our research to related areas and reviewed al types of scientific studies. Therefore, we used mainly open access texts from the last five years, but we also included standard authors related to our type and considered older scientific investigations.

* Which were the Exclusion criteria during the review. Please see below some examples: Not belong to the field of research, Non English, Publication date before….., Non scientific studies, Not related to the subject of interest.

We excluded articles that included external influences over food production systems, because we concentrated on the food production systems on themselves. Moreover, we investigated in different languages so we could have a more amplified few on the topic and evolution about the last years. 

* The authors need to perform also a detailed language editing, since sometimes Spanish words and phrases appear in the text.

We have evaluated the situation of the English quality of our article and we have decided to send the article for further editing to native English speakers so that we can have a better English quality of our manuscript. Furthermore, we changed all words and phrases that appeared in Spanish to English.

Round 2

Reviewer 1 Report

Comments to the authors:

The present study describes the sustainability pillars for food production in order to improve the sustainable food production and food security. The authors have proposed five sustainability pillars, which are not well connected to sustainable food production and food security. I suggest, it needs a connection to the present study rather generalizing each pillar. Otherwise, the papers are divided into two parts, firstly sustainability pillars and secondly food production system, which does not form a precise connection.

The quality of the paper and language have improved now. Though, the Ms still lack a precise connectivity.

Specific comments:

Method section- The parameters of systematic review are not followed, it needs to follow the clear cut method of systematic review.

L-127 Correct the highlighted portion.

Add the heading ‘Results’ in the text after method section.

The English language has improved now.

Author Response

Reviewer 1

* The present study describes the sustainability pillars for food production in order to improve the sustainable food production and food security. The authors have proposed five sustainability pillars, which are not well connected to sustainable food production and food security. I suggest, it needs a connection to the present study rather generalizing each pillar. Otherwise, the papers are divided into two parts, firstly sustainability pillars and secondly food production system, which does not form a precise connection.

Thank you for your comments.

We think that we have well connected the traditional pillars of sustainability with the proposed separation of culture from the social pillar and the separation of technology from the economic pillar to explain in a better way how economics, social aspects, environmental behavior, cultural heritage and technology impact food production in different systems. Therefore, we made this division in our article to explain why it is important to define culture and technology as separated sustainable pillars, which helps us to explain how this pillars impact food production and what practices or characteristics of a food production systems are sustainable and which ones are not.

* The quality of the paper and language have improved now. Though, the Ms still lack a precise connectivity.

We checked the document you attached to your comment, reviewed our article, and adopted a lot of your recommendations to improve the connectivity of our article. We marked the parts we adopted from your recommendations yellow and the parts we think from our version is more accurate in green. We hope that our article is more readable after these changes and that the understanding of our topic was improved. 

Specific comments:

* Method section- The parameters of systematic review are not followed, it needs to follow the clear cut method of systematic review.

Original: “The initial literature research in June 2022 was established to provide basic information about the concept of sustainability in relation with food production, which provided the keywords for the review. The research was performed by searching the MDPI, El Servier, IEEE, Wiley, Taylor & Francis, and Google Scholar databases, concentrating the research on the sustainability dimensions and food production systems. This differentiation helped to define a methodology with two main sections: analyzing the need to complement the traditional pillars of sustainability in food production and the aspects of cultural heritage and technological production impacts, as well as the re-flection of these predefined sustainable pillars in different food production systems.

Therefore, we performed a systematic review of the three traditional pillars of sustainability; economic, social and environmental to analysis the different concepts and implement the following sub-questions:

  1. What are the details of the traditional sustainable dimensions?
  2. Is there a need to amplify this definition based on food production systems?
  3. How sustainable are traditional food production systems (agriculture, aquaculture) are?
  4. How sustainable is an aquaponic production system in comparation to the traditional production systems?

We attempt to answer these sub-questions by; 1. describing the traditional dimensions of sustainability; 2. based on the traditional production systems, we ubicated culture and technology need to be more specific defined to implement a better under-standing of sustainable food production; 3. we analyzed two traditional food production systems, agriculture and aquaculture and their sustainable impact in food production and compered these systems 4. to sustainable aquaponic food production. 

Although aquaponics is categorized as a sustainable production method, we want to compare the sustainability of aquaponic food production with the traditional pro-duction system, because aquaponics is the combination of agriculture and aquaculture and both systems have some difficulties to accomplish full sustainability.”

Modified (marked in blue): “The initial literature research in June 2022 was established to provide basic information about the concept of sustainability in relation with food production, which provided the keywords for the review. The research was performed by searching the MDPI, El Servier, IEEE, Wiley, Taylor & Francis, and Google Scholar databases, concentrating the research on the sustainability dimensions and food production systems. This differentiation helped to define a methodology with two main sections: analyzing the need to complement the traditional pillars of sustainability in food production and the aspects of cultural heritage and technological production impacts, as well as the re-flection of these predefined sustainable pillars in different food production systems.

Therefore, we performed a systematic review of the three traditional pillars of sustainability; economic, social and environmental to analysis the different concepts and implement the following sub-questions:

  1. What are the details of the traditional sustainable dimensions?
  2. Is there a need to amplify this definition based on food production systems?
  3. How sustainable are traditional food production systems (agriculture, aquaculture) are?
  4. How sustainable is an aquaponic production system in comparation to the traditional production systems?

We attempt to answer these sub-questions by; 1. describing the traditional dimensions of sustainability; 2. based on the traditional production systems, we ubicated culture and technology need to be more specific defined to implement a better under-standing of sustainable food production; 3. we analyzed two traditional food production systems, agriculture and aquaculture and their sustainable impact in food production and compered these systems 4. to sustainable aquaponic food production. 

The analysis of the traditional sustainable dimensions was performed reviewing the initial concept of sustainability and the adoption by international institutions during the last decades. Moreover, the analysis of the two traditional production systems (agriculture and aquaculture) showed that there are cultural and technological differences globally that need to be considered to define sustainable food production. Furthermore, food production was analyzed on the base of external influences which substituted the traditional production methods, like the case of aquaponic food production.

Although aquaponics is categorized as a sustainable production method, we want to compare the sustainability of aquaponic food production with the traditional pro-duction systems, because aquaponics is the combination of agriculture and aquaculture, and both systems have some difficulties to accomplish full sustainability.

This research was performed in three different languages; English, German and Spanish to gain different information and studies about the sustainable pillars. Moreover, the research in different languages helped to review articles from different countries and to understand the cultural heritage of food production and applied technology in different parts of the world.

For this research we utilized different types of articles (review and scientific studies) and book chapters from different areas where sustainable pillars are applied and defined, and about sustainability in food production systems. In general, the information is dated from 2018 to 2023, only established concepts and basic information about the topic is dated from before 2018.”

* L-127 Correct the highlighted portion.

Original: “Therefore, the following part concentrates on the five principal sustainability pillars (Figure 1.) and their classification.”

Modified: “Therefore, the following part concentrates on the three principal sustainability pillars, their classification, and the importance of the definition of the cultural and technological pillars of sustainability (Figure 1.).” 

* Add the heading ‘Results’ in the text after method section.

We decided to structure our article in this format because it was not designed for results form an experiment rather than for a review of sustainability and the application in the article mentioned food production systems. So, we decided to keep the current version “The concept of sustainability and the sustainable pillars” because we think it is more accurate for this review article due to the number of subtitles we implemented in this section. Furthermore, a format change would require major corrections in the article, because the discussion is also adopted to the current format. Also, we checked other review articles in MDPI´s Sustainability Journal and there are published articles that have the format you propose, but on the same time there are also a lot of articles published in the format we chose.    

Reviewer 3 Report

Thanks to the authors for improving the manuscript according to the reviewer's comments.

Author Response

No comments were done

Reviewer 4 Report

Dear authors

Thank you very much for proceeding with the recommended corrections.

Your article has been greatly improved.

Author Response

No comments were done

Round 3

Reviewer 1 Report

L-120  sustainable pillars or sustainability pillars?

Author Response

Reviewer 1

* L-120  sustainable pillars or sustainability pillars?

Original: “This research was performed in three different languages; English, German and Spanish to gain different information and studies about the sustainable pillars.”

Modified: “This research was performed in three different languages; English, German and Spanish to gain different information and studies about the pillars of sustainability”

Thank you very much for your comments.